# Effect of Radio-Chemotherapy on PD-L1 Immunohistochemical Expression in Head and Neck Squamous Cell Carcinoma

**DOI:** 10.3390/jpm13020363

**Published:** 2023-02-18

**Authors:** Ilaria Girolami, Stefano Marletta, Vincenzo Fiorentino, Simonetta Battocchio, Bruna Cerbelli, Barbara Fiamengo, Clara Gerosa, Andrea Gianatti, Luca Morelli, Giulio Riva, Maria Giovanna Zagami, Nicola Fusco, Enrico Munari, Vincenzo L’Imperio, Fabio Pagni, Patrizia Morbini, Maurizio Martini, Albino Eccher

**Affiliations:** 1Department of Pathology, Provincial Hospital of Bolzano (SABES-ASDAA), Lehrkrankenhaus der Paracelsus Medizinischen Privatuniversität, 39100 Bolzano-Bozen, Italy; 2Department of Diagnostics and Public Health, University of Verona, 37124 Verona, Italy; 3Pathology Unit, Pederzoli Hospital, 37019 Peschiera del Garda, Italy; 4Division of Anatomic Pathology and Histology, Fondazione Policlinico Universitario A. Gemelli-IRCCS, 00168 Rome, Italy; 5Department of Pathology, ASST Spedali Civili, University of Brescia, 25123 Brescia, Italy; 6Department of Medico-Surgical Sciences and Biotechnology, Sapienza University of Rome and Policlinico Umberto I, 00161 Rome, Italy; 7Department of Pathology, Humanitas Clinical and Research Center Istituti di Ricovero e Cura a Carattere Scientifico (IRCCS), 20089 Milan, Italy; 8Department of Pathology, San Giovanni di Dio Hospital, University and Hospital Trust of Cagliari, 09124 Cagliari, Italy; 9Pathology Unit, ASST Papa Giovanni XXIII, 24127 Bergamo, Italy; 10Pathology Unit, Santa Chiara Hospital, 38122 Trento, Italy; 11Pathology Unit, San Bortolo Hospital, 36100 Vicenza, Italy; 12Pathology Unit, Siracusa Hospital, 96100 Siracusa, Italy; 13Department of Oncology and Hemato-Oncology, University of Milan, 20141 Milan, Italy; 14Pathology Department, San Gerardo Hospital, Department of Medicine and Surgery, University of Milan-Bicocca, 20900 Monza, Italy; 15Unit of Pathology, IRCCS Fondazione Policlinico San Matteo, University of Pavia, 27100 Pavia, Italy; 16Division of Anatomic Pathology and Histology, University of Messina, 98124 Messina, Italy; 17Department of Pathology and Diagnostics, University and Hospital Trust of Verona, 37126 Verona, Italy

**Keywords:** programmed death-ligand 1, head and neck squamous cell carcinoma, immunotherapy, immunohistochemistry, radio-chemotherapy, systematic review

## Abstract

Background: Programmed death-ligand 1 (PD-L1) checkpoint inhibitors represent a mainstay of therapy in head and neck squamous cell cancer (HNSCC). However, little is known about the influence of combined therapy on PD-L1 expression. The study aims to gather evidence on this topic. Methods: A systematic search was carried out in electronic databases Pubmed-MEDLINE and Embase to retrieve studies on the comparison of PD-L1 expression before and after conventional therapy. Data were extracted and a quantitative analysis with pooled odds ratios (ORs) was performed when applicable. Results: Of 5688 items, 15 were finally included. Only a minority of studies assessed PD-L1 with the recommended combined positive score (CPS). The results are highly heterogeneous, with some studies reporting an increase in PD-L1 expression and others reporting a decrease. Three studies allowed for quantitative analysis and showed a pooled OR of 0.49 (CI 0.27–0.90). Conclusions: From the present evidence, a clear conclusion towards an increase or decrease in PD-L1 expression after combined therapy cannot be drawn, but even with few studies available, a trend towards an increase in expression in tumor cells at a cutoff of 1% can be noted in patients undergoing platinum-based therapy. Future studies will provide more robust data on the effect of combined therapy on PD-L1 expression.

## 1. Introduction

Head and neck squamous cell carcinoma (HNSCC) represents a leading cause of mortality in some countries, affecting approximately 880,000 new patients each year worldwide [1]. Despite combined therapy with surgical resection plus radiotherapy and/or chemotherapy, the 5-year overall survival has improved only modestly over the past three decades, and it is only 50–65% [2,3]. The introduction of immunotherapy targeting the programmed death-1 (PD-1)/programmed death-ligand-1 (PD-L1) axis has represented a turning point in the therapeutic management of HNSCC [4,5,6,7]. The US Food and Drug Administration (FDA) and European Medicines Agency (EMA) approved the use of PD-1 inhibitors pembrolizumab and nivolumab for recurrent and metastatic disease, given that these drugs improved survival with reduced toxicity [8,9,10,11]. In clinical trials, simple immunohistochemistry (IHC) has been deployed to assess PD-L1 expression in tumor and immune cells related to therapeutic response, with several scoring systems [12]. The main scoring systems to evaluate the expression of PD-L1 are the tumor proportion score (TPS), defined as the proportion of tumor cells staining positive at any intensity with the marker, and the combined positive score (CPSs), which also take into account the positivity of the immune cells closely associated with the tumor, thus being defined as the ratio of total positive tumor and immune cells to the total number of viable tumor cells. The KEYNOTE-040 study reported a significant survival benefit in patients with recurrent or metastatic HNSCC treated with pembrolizumab whose tumor biopsies showed a PD-L1 expression higher than 50% with TPS [4]. The KEYNOTE-048 study deployed CPS and showed that its best predictive performance is achieved when the cutoff of CPS is ≥20, while a lower but significant improvement in survival is present also at CPS ≥1 for first-line treatment [5]. Moreover, a subsequent post hoc analysis highlighted an equivalence of CPS ≥ 50 to TPS ≥ 50% for the prediction of survival outcome measures in HNSCC patients [13]. Differences in the prediction of therapeutic response with the two scoring systems emerged from trials; indeed, the inclusion of immune cells in the CPS leads to an overall average PD-L1 positivity of HNSCC of 85%, while it is around 50–60% when assessed with TPS [12,14]. Consequently, CPS sensitivity appears to be higher even at lower cutoffs, supporting the importance of PD-L1-positive immune cells [13]. Eventually, a CPS of ≥1 was indicated by international agencies as a selection criterion for first-line treatment of recurrent and metastatic HNSCC with pembrolizumab.

At present, the use of immunotherapy with PD-L1 checkpoint inhibitors has increased greatly and pathologists are even more often required to assess PD-L1 expression in patients’ specimens. Given that IHC expression with CPS remains the prerequisite for immunotherapy administration, it is of preeminent importance to know the factors influencing PD-L1 expression, ultimately conditioning access to drugs. Most of the published studies focused on the reliability and interchangeability of IHC clones and platforms for PD-L1 evaluation, showing variable degrees of concordance among the 22C3 reference clone and the other clones used [15,16,17,18,19]. Apart from differences among clones, some studies then investigated the potential differences in PD-L1 expression between different specimens, with a focus on concordance between biopsy and final resection specimen or between expression in primary tumor and subsequent lymph node metastases [20,21,22,23,24]. Moreover, recent evidence highlights a significant decrease in PD-L1 expression in HNSCC specimens with time [25]. However, little is known as of yet on the influence of conventional therapies, e.g., radiation and chemotherapy regimens, on PD-L1 expression. This is important because PD-L1 checkpoint inhibitors are generally administered in patients who have already undergone other therapies unsuccessfully, and, thus, represent the last therapeutic option. This work aims to systematically gather evidence from the published literature on the impact of conventional therapies on PD-L1 expression and discuss their potential implications.

## 2. Materials and Methods

A search query comprising the key terms “PD-L1” and “HNSCC” with all their aliases was run on the literature databases Pubmed-MEDLINE and Embase until 12 June 2022 to provide a systematic literature search. The complete search strategy is found in Appendix A. The review question was modeled on a Population, Index/Intervention, Comparator, Outcome (PICO) model, where the Population was represented by HNSCC patients whose biopsy or surgical material was tested for PD-L1 expression; Index was the expression of PD-L1 after therapy, Comparator was the expression of PD-L1 before therapy, and the Outcome was simply the difference in expression, reported in any measure. Therefore, inclusion criteria were the presence of any type of comparison of the PD-L1 expression with IHC before and after any type of therapy in HNSCC patients. No language restrictions were applied. Studies not dealing with HNSCC, not dealing with PD-L1 assessment in IHC, or not presenting any type of comparison were excluded, as well as studies represented by abstracts only, letters, reviews, and editorials with no comparison data. Abstracts retrieved were screened by a group of authors with the aid of the Rayyan web application [26]. Selected studies were then assessed in full-text form to decide on final inclusion and data extraction. Data extracted were: author, year and country of study, number of cases with basic demographic data of patients, subsite of HNSCC, IHC clone and platform used, scoring system used, main results on the effect of therapy, and additional results and limitations. Data were extracted on a shared Excel spreadsheet by the same group of authors and reviewed subsequently for disagreement and final approbation by two other authors (IG, SM). When studies provided raw data regarding prevalence of PD-L1 positivity in HNSCC specimens before and after conventional therapy, a comparison meta-analysis was performed. High heterogeneity was expected in terms of clones and platform used, scoring system deployed, and specific type of therapy. Therefore, we chose to perform a meta-analytical comparison only when studies used the same clone and provided raw data on the proportion of positive cases separately in tumor and immune cells, with the same 1% cutoff for positivity. Results were expressed as an odds ratio (OR) with positivity for PD-L1 as outcome event, and pooled OR with 95% CIs was calculated using DerSimonian–Laird fixed-effects model. Heterogeneity across studies was assessed by the I^2^ metric and chi-square statistics [27,28]. All analyses were carried out using Review Manager 5.3 (The Nordic Cochrane Center, Cochrane Collaboration, Copenhagen, 2014). The quality of the evidence was evaluated with a GRADE approach, as advised for systematic reviews with meta-analysis [29].

## 3. Results

Of 5688 items after duplicate removal, 15 were finally included (Figure 1).

Overall, the studies report on a total of 920 patients (range 21–161), with studies coming from Europe (*n* = 6, 40%) [30,31,32,33,34,35], China (*n* = 3, 20%) [36,37,38], Japan (*n* = 3, 20%) [39,40,41], Korea (*n* = 2, 13%) [42,43], and the USA (*n* = 1, 7%) [44]. The studies were published in the time range 2017–2021, with the vast majority appearing in the years 2020 and 2021. The tumors were located in most studies in more than one subsite of the head and neck region, with some studies dealing with nasopharyngeal cancer only [36,37], with tongue only [39], or with hypopharynx only [38]. The main clones used were E1L3N in six studies (40%) and SP142 in two studies, while clones SP263, 28-8, QR1 from Quartett, D3 from Ventana, ab156361 from Abcam, and 66248-1-Ig from Proteintech were used in one study each; a study used multiplex immunohistochemistry but not stated clones. The recommended scoring system with CPS was reported in three studies (20%), while the majority of studies either used TPS or scored tumor cells and immune cells separately. The conventional therapies administrated to patients consisted, in most of the studies, of platinum-based regimens with concurrent radiation therapy, but there were two studies that investigated the effect of other types of chemotherapy, such as a combination of cytokine preparation [44] and an inhibitor of apoptosis protein injection [32]. A study reported on the effect of radiotherapy only [43]. The period between the start or the completion of therapy and the subsequent testing of PD-L1 was available only in 3 out of 15 studies, which reported 8 weeks [36] or median times of 3.7 months [31] and 11.6 months [42]. A summary of included studies listed by IHC clone and their main results is reported in Table 1, while more detailed data on the studies are reported in Appendix A.

Concerning the effect of therapy on PD-L1 expression, results from studies are highly heterogeneous. There were reports of a decrease in PD-L1 expression with concurrent radiotherapy and platinum-based chemotherapy, as well as reports of an increase in expression. Three studies deployed clone E1L3N, scored tumor and immune cells separately with the same cutoff at 1% for positivity, and provided raw data on the proportion of positive cases to allow for quantitative comparison [30,34,42]. We found a pooled OR of 0.49 (95% CI 0.27–0.91, *p* = 0.02), with moderate residual heterogeneity, showing a significant increase in PD-L1 expression after combined chemo-radiotherapy (Figure 2). The same comparison for immune cells did not show significance (data not shown).

## 4. Discussion

PD-L1 checkpoint inhibitors represent, nowadays, a mainstay of therapy in HNSCC. Whilst PD-L1 expression has also shown variable prognostic value in several cancers, with conflicting evidence [45,46,47,48,49,50], its expression assessed with CPS is the prerequisite for the administration of therapy, and awareness and control of factors that can influence PD-L1 expression are of preeminent importance to avoid misinterpretation of this marker. In this setting, little is known about the effect of conventional combined regimens with chemo- and radiotherapy.

Our systematic review is interested in this specific topic and found a total of 15 studies investigating the issue. The first result is that there is high heterogeneity in studies regarding the clone used, the scoring system, and the therapy regimens. Indeed, only a minority of studies deployed the CPS, as indicated in trials and agency approvals, and there was also great variability in clones and platforms used, with only a minority of studies deploying approved clones, such as 22C3, SP263, or SP142. This is important to notice because it is known that evaluating HNSCC with CPS yields a higher quota of positive—and, thus, eligible for therapy—cases in comparison to TPS or scoring tumor cells alone [12]. Even in studies where the tumor and immune cells were scored separately, a combined score was not provided. We can only speculate on the reason for this, given that most of the studies were published after the FDA approval in 2019 with the indication of cutoff CPS ≥ 1. It could be considered that assessing PD-L1 expression separately for tumor and immune cells may be easier for pathologists involved in studies. Indeed, results from studies were highly heterogeneous, with no clear trend of an increase or decrease in expression. Studies that compared expression in tumor and immune cells with a statistical test, in some cases, did not find a significant change in PD-L1 expression [31,40], while in other studies, a significant increase in the quota of positivity was reported [34,42]. The study of So et al. reported on the effect of radiotherapy, even though there was a minor quota of patients also treated with chemotherapy, and showed a decrease in PD-L1 expression in 69% of cases, with negativization in 23% [43]. In one study, the effect of a precursor of fluorouracil alone was investigated and the authors reported a significant decrease in PD-L1 expression in tumor cells [41], while the other studies where fluorouracil was deployed reported, on the contrary, an increase in the quota of positive cases [34,42]. In these studies, however, fluorouracil was used in combination with a platinum agent and docetaxel, and results were not stratified for different combinations of chemotherapy or cisplatin-based therapy separately. Moreover, it must be observed that in most of the studies, radiation therapy was concurrent to chemotherapy, and no studies reported the results of the minority of patients in their population who received radiotherapy alone. Moreover, the time between the start or completion of therapy and the subsequent IHC testing of PD-L1 was not reported in most studies. Indeed, the time of testing is important, as recent evidence has shown that there is a decrease in PD-L1 expression in specimens of HNSCC with time, and that an immediate testing rather than a retrospective testing is advised [25]. For many studies, we do not have the precise timing of PD-L1 testing, because whilst the post-therapy specimens are reasonably tested immediately, some of the before-therapy specimens could have been tested in a subsequent time for the purposes of the study, with no control on the real ageing of the specimen and the potential biasing effect on PD-L1 expression. Therefore, we must conclude that from present evidence, a clear trend towards an increase or decrease in PD-L1 expression after combined therapy cannot be drawn. This is, however, only partly surprising, given the intrinsic complexity of HNSCC and its relation with the tumoral microenvironment (TME). Recent evidence suggests that many other markers and mechanisms are involved in the dysregulation of the cancer-related immune system and this could explain non only the limited quota of effective responses to immunotherapy [12] but also the differences in expression of related markers such as PD-L1 [51]. Indeed, the PD-1/PD-L1 axis is not the only relevant mechanism of immune evasion. Other molecules, such as TIGIT and FOXP3, have been investigated as regulatory key points of the TME and, therefore, as potential targets of new immunotherapeutic drugs [52,53], and so-called tertiary lymphoid structures, as shown to be important in prediction of the outcome in HNSCC [54]. Moreover, the presence of HPV-related carcinogenesis and differences in oral cavity microbiota have been shown to influence the composition of the TME and the interactions among cells [55]. Therefore, it emerges that it is not just one immune player that must be targeted but, rather, an entire orchestra of immune and non-immune players for robust improvements in therapeutic strategies for patients with HNSCC tumors [51]. In light of this, we have to be aware that differences in PD-L1 expression among patients—at present, the only marker corresponding to an available and effective drug—could be influenced by the interactions of this plethora of elements in the immune TME, and, consequently, the apparent effect of combined therapy could be the result of the effect of therapy on this complex environment.

To partly overcome the heterogeneity, we tried to find, among all the included studies, a subset of studies with sufficient homogeneity in terms of clone used, cutoff, and scoring method to perform a quantitative analysis. Three studies deployed clone E1L3N, and all separately scored tumor and immune cells with the same cutoff at 1% for positivity and provided raw data on the proportion of positive cases to allow for quantitative comparison [30,34,42]. We found a pooled OR of 0.49 (95% CI 0.27–0.91), with moderate residual heterogeneity, showing a significant increase in PD-L1 expression after therapy in tumor cells alone. Moreover, these studies shared the presence of a platinum agent in their therapy regimens and, in two of them, there was also the administration of concurrent radiotherapy. This is in line with other studies reporting the effect of cisplatin as an enhancer of PD-L1 expression on squamous and non-squamous cell cancer in other organs [56,57,58,59,60,61], and our result may be the first quantitative evaluation of the effect of therapy schemes with a platinum agent on PD-L1 expression. However, we are aware that the number of studies is low and the total number of patients is also very low, thus implying low quality in the evidence, according to a GRADE system evaluation scheme [29]. In our case, we had all retrospective studies with a low total number of patients/cases, so even though there was no inconsistency in the results of studies included in the meta-analysis, the overall quality has to be considered low. At the same time, we are also persuaded that this result may represent an important hint in the determination of the effects of chemotherapy on PD-L1 expression. Taking all these findings together, it should be reasonable to retest PD-L1 expression in patients who have undergone conventional therapy, to be sure not to miss a positivization, which would imply access to therapy for the patient, nor a negativization, thus preventing inappropriate drug administration. From a clinical perspective, there are, therefore, many issues to take into account: the intrinsic variability in PD-L1 expression in cancer population, the potential bias due to aging of the specimens—hence, the practice in some institutions to test patients for PD-L1 at first presentation, irrespective of expected outcome or clinical request, which, however, raises important questions on economic resource use—and, finally, the potential effect of already administered therapy on PD-L1 expression and, thus, eligibility for immunotherapy. Obviously, this issue has to be discussed in the setting of a multidisciplinary approach to the patient, involving not only the pathologists who are responsible for the reliability and accuracy of PD-L1 assessment and handling of the specimens, but also the clinicians that can evaluate whether the patient could tolerate eventual re-biopsy if needed to gain more recent material.

Our study has some strengths and some limitations. The strengths reside in the systematic approach to the review question and the acknowledgment of the high heterogeneity in the studies included. Indeed, we chose to be conservative and attempt a quantitative analysis only on studies with sufficient homogeneity in terms of clone used, scoring method, and cutoff used. The limitations pertain to the limited number of studies and patients analyzed, given that the majority of studies did not report separately on the effect of different therapy regimens nor adhere to the recommended scoring system, and, therefore, the final quality of evidence gathered is to be considered low. However, the study highlighted some points, which can be taken into account in future studies. Indeed, the need for use of the recommended CPS and of a validated clone, the control of the timing of assessment of PD-L1 to take into account the age of the specimens, and the more precise definition of therapy regimens with a stratified analysis will help to better evaluate the impact of a single agent or a specific combination of drugs on the expression of PD-L1.

## 5. Conclusions

Evidence on the effect of combined therapy on PD-L1 expression in HNSCC is still immature and studies on this topic reported conflicting results. Most of them did not use CPS nor a validated clone, and this may have biased the results in terms of the quota of positive and negative cases. A trend toward an increase in expression in tumor cells at a cutoff of 1% can be noted in patients undergoing combined therapy with a platinum agent.

## Figures and Tables

**Figure 1 jpm-13-00363-f001:**
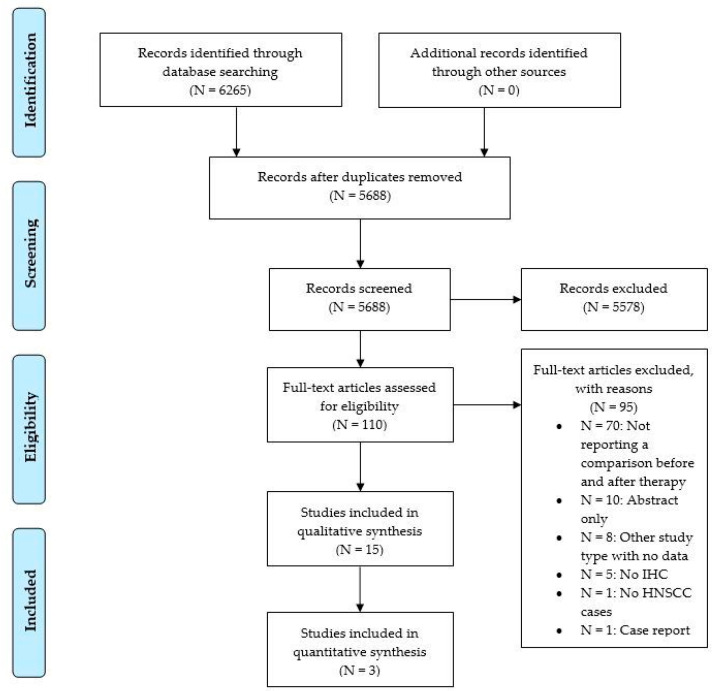
Flow chart of article screening according to PRISMA.

**Figure 2 jpm-13-00363-f002:**
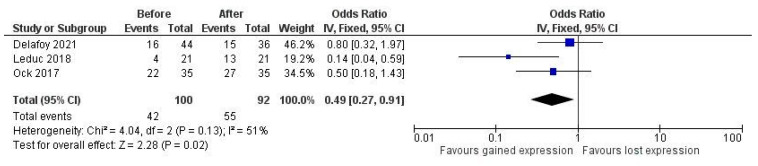
Pooled odds ratio (OR) of PD-L1 expression after therapy.

**Table 1 jpm-13-00363-t001:** Summary of retrieved studies investigating the impact of chemo- and radiotherapy on PD-L1 expression in head and neck squamous cell carcinoma.

Author, Year (Country)	N Cases	Sites	Therapy	Scoring System	Clone	MAIN RESULTS	Potential Limitations
Clone E1L3N (6 studies)							
Delafoy, 2021 (France) [30]	44	HNSCC NOS	Concurrent cisplatin and RT	TC and IC scored separately, positive if >1%	E1L3N	Positivization in TC in the majority of cases where PD-L1 expression changed after therapy, but decrease after radiation	No correspondence between Bx and Rx nor separation of data
Doescher, 2020 (Germany) [31]	67	HNSCC excluded sinonasal and RP	Concurrent platinum-based CHT and RT	TC and IC scored separately with H-score	E1L3N	No significant changes in the expression of PD-1 on neither tumor nor TIL during therapy	No use of established CPS cutoffs as in trials, small sample size, heterogeneity in tumor locations, fixation and storage of tissue blocks and treatments with no stratification
Gomez-Roca, 2020 (France) [32]	26	Oral, OP, L, HP	Inhibitor of apoptosis protein Debio 1143 with and without cisplatin	TC and IC as % positive cells	E1L3N	Increased PD-L1 levels in IC with neoadjuvant target therapy with inhibitor of apoptose protein and in combination with cisplatin, but not with cisplatin alone, not in TC	Pharmacodynamic and pharmacogenomic study with no use of established CPS cutoffs as in trials
Leduc, 2018 (France) [34]	21	Oral and L	Docetaxel, platinum and fluorouracil (TPF)	TC and IC scored separately with >5% as positive	E1L3N	Significant increase of PD-L1 expression in IC and TC after therapy	Small sample size
Long, 2021 (China) [37]	24	NP	Concurrent IMRT and cisplatin	TPS, positive if >1%, CPS	E1L3N	Different distribution of changes in PD-L1 expression according to score used whether TPS or CPS	Not full correspondence of data in tables and text of the article
Ock, 2017 (Korea) [42]	35	OP mainly	Docetaxel, platinum and fluorouracil (TPF) regimens or cisplatin-based regimens and concurrent RT	TPS > 5% as positive	E1L3N	Statistically significant up-regulation of PD-L1 in 69% of originally negative cases and upregulation of PD-L1 after radiotherapy	Small sample size, and unclear detailing of IHC protocol
Clone SP142 (2 studies)							
Chan, 2017 (China) [36]	161	NP	IMRT and CHT with cisplatin	TC and IC with cutoff 1%	SP142	Reduction of PD-L1 expression in 56% cases in IC and in 33% in TC after treatment	No separated data for patients receiving radiotherapy alone
So, 2020 (Korea) [43]	42	Oral cavity, L, OP, HP, sinuses	RT only	TC and IC positive if >10%, automated scoring on WSI digitized slides	SP142	Ratio of recurrent (R) vs. initial (I) PD-L1 expression was <1 in 69% cases with TC turning negative in 23% and positive in 9%	Small sample size, retrospective design, and different treatment characteristics among patients
Other clones (7 studies, a clone each)							
Karabajakian, 2021 (France) [33]	35	Oral, OP, HP, L	RT-CHT NOS	CPS	QR1	Majority of samples positive at CPS1 remained positive (76%), while majority of negative CPS samples at diagnosis became positive at relapse (75%), but change of different entity at different cutoffs CPS1 and CPS20	No special limitations
Naruse, 2020 (Japan) [39]	121	Tongue	Platinum-based CHT	TC only, positive if >5%	Abcam ab156361	No marked difference in expression of PD-L1 in recurrent patients who had surgery alone or CHT plus surgery	No use of CPS or approved assay and no reporting of CI or *p* values for comparisons; only reporting for correlation with clinicopathological variables and prognosis
Ono, 2020 (Japan) [40]	30	L, OP, HP	Cisplatin-based RT-CHT	TPS and IC density score	clone D3	No significant changes in PD-L1 expression after RT-CHT	Small sample size
Pflumio, 2021 (France) [35]	100	Oral, OP, L, HP	RT-CHT NOS	TPS > 1%	SP263	Significantly lower percentage of PD-L1+ TC within the irradiated area cohort than the de novo cohort and significantly fewer tumors with PD-L1+ IC in the irradiated area cohort	Indirect comparison, as 50 irradiated cancers are compared with 50 de novo cancers and no CPS used
Seki-Soda, 2021 (Japan) [41]	71	Oral cavity	Preoperative fluorouracil-based CHT	TC	28-8	PD-L1 positivity in TC significantly decreased by CHT irrespective of clinical response	No use of established CPS cutoffs as in trials
Shen, 2020 (China) [38]	47 age 64 y)	HP	Platinum-based CHT and RT	TC	66248-1-Ig	PD-L1 expression increased after RT	Not stated a clear cutoff for positivity, only 10% for low vs. high
Wolf, 2020 (USA) [44]	96	Oral cavity	Multi-cytokine preparation with cyclophosphamide	IC and H-score, CPS	Multiplex fluorescence IHC	No significant changes in PD-L1 expression after CHT	Small sample size

Bx, biopsy specimen; CHT, chemotherapy; CI, confidence interval; CPS, combined positive score; EMT, epithelia-mesenchymal transition; FU, follow-up; HNSCC, head and neck squamous cell carcinoma; HP, hypopharynx; IC, immune cells; IHC, immunohistochemistry; IMRT, intensity modulated radiation therapy; L, laryngeal; M, male; NOS, not otherwise specified; NP, nasopharyngeal; OP, oropharyngeal; OS, overall survival; PFS, progression-free survival; RP, rhinopharynx; RT-CHT, radio-chemotherapy; Rx, resection specimen; TC, tumor cells; TILs, tumor-infiltrating lymphocytes; TPS, tumor proportion score; WSI, whole-slide imaging; y, years.

## Data Availability

Not applicable.

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
