# Peer review of "Effect of Radio-Chemotherapy on PD-L1 Immunohistochemical Expression in Head and Neck Squamous Cell Carcinoma"

_jpm, 2023, doi:10.3390/jpm13020363_

Round 1

Reviewer 1 Report

Dear authors,

I enjoy reading your work. I like your Introduction pinpointing the most imperative information. Throughout this mini-review, the authors summarize and outline key terms to help readers to comprehend and organize better. Current evidence on effect of combined therapy on PD-L1 expression in HNSCC is still conflicting. I’m impressed by this article from a research study dedicated to clarify this issue. 

1. Please comply with our journal’s regulation. 

Manuscripts submitted to JPM should 

https://www.mdpi.com/journal/jpm/instructions

Review: Reviews offer a comprehensive analysis of the existing literature within a field of study, identifying current gaps or problems. They should be critical and constructive and provide recommendations for future research. No new, unpublished data should be presented. The structure can include an Abstract, Keywords, Introduction, Relevant Sections, Discussion, Conclusions, and Future Directions, with a suggested minimum word count of 4000 words.  Please specify your word count.

2. Acronyms/Abbreviations/Initialisms should be defined the first time they appear in each of three sections: the abstract; the main text; the first figure or table. When defined for the first time, the acronym/abbreviation/initialism should be added in parentheses after the written-out form.

Line 43, CPS?

3. Please list “radio-chemotherapy “ (as the term in your title) in keyword for more citation of your ork.

4. Please explain the credibility of the utilization of Rayyan in this research study.

5. Table 1 is plausible. Please list ref. number after author, year for better readability. 

6. Figure 2 is blurry. Please refer to 

https://www.mdpi.com/journal/jpm/instructions#figures

Author Response

Dear authors,

I enjoy reading your work. I like your Introduction pinpointing the most imperative information. Throughout this mini-review, the authors summarize and outline key terms to help readers to comprehend and organize better. Current evidence on effect of combined therapy on PD-L1 expression in HNSCC is still conflicting. I’m impressed by this article from a research study dedicated to clarify this issue. 

We thank the Reviewer for appreciating our work.

Point 1: Please comply with our journal’s regulation. 

Manuscripts submitted to JPM should 

https://www.mdpi.com/journal/jpm/instructions

Review: Reviews offer a comprehensive analysis of the existing literature within a field of study, identifying current gaps or problems. They should be critical and constructive and provide recommendations for future research. No new, unpublished data should be presented. The structure can include an Abstract, Keywords, Introduction, Relevant Sections, Discussion, Conclusions, and Future Directions, with a suggested minimum word count of 4000 words.  Please specify your word count.

Response 1: We thank the Reviewer for the suggestions. We checked the journal’s guidelines and added the word count, which is 4216 words comprising text, tables and legends and excluding title page, abstract, back matter and references.

Point 2: Acronyms/Abbreviations/Initialisms should be defined the first time they appear in each of three sections: the abstract; the main text; the first figure or table. When defined for the first time, the acronym/abbreviation/initialism should be added in parentheses after the written-out form.

Line 43, CPS?

Response 2: We checked the acronyms and corrected when they first appeared in abstract and text.

Point 3: Please list “radio-chemotherapy “ (as the term in your title) in keyword for more citation of your ork.

Response 3: We thank the Reviewer for the suggestion. We modified the keyords’ list accordingly.

Point 4: Please explain the credibility of the utilization of Rayyan in this research study.

Response 4: We thank the Reviewer for the question. Rayyan is a web-based application for the organization of screening of titles and abstract for systematic reviews, it offers an automatic function of deduplication, and has a free version (differently from other wide-spread softwares like Covidence) which has all the essential functionalities for this step of the systematic review process, and its popularity and use is constantly increasing. We added a brief note on its use in the Methods section with referencing.

Point 5: Table 1 is plausible. Please list ref. number after author, year for better readability. 

Response 5: We thank the Reviewer for the suggestion. We added the references number in the Table.

Point 6: Figure 2 is blurry. Please refer to 

https://www.mdpi.com/journal/jpm/instructions#figures

Response 6: We thank the Reviewer for the suggestion. We checked and modified the format of the Figure 2, which now is .jpeg and with a resulution of 300dpi as for the instructions.

Reviewer 2 Report

The authors evaluated the changes in PD-L1 expression levels in IHC before and after conventional cancer therapy in head and neck SCC by database systematic search.

This is an attractive perspective because, in clinical practice, PD-L1 assessment directly affects the selection of cancer therapy.

However, it should be drastically updated for publication. 

This manuscript is partially difficult to read due to immaturity in English language expression and sentence structure. Specifically, the introduction is redundant, there is some overlap between the discussion and results parts, and there are errors in the refs (e.g., #8). 

Fig 1 is very confusing to understand and needs a significant layout change. The order of authors' names is meaningless. It is preferable to list by treatment method or by clone. It would also be informative to specify the period between the start (or completion) of treatment and the next IHC.

Author Response

The authors evaluated the changes in PD-L1 expression levels in IHC before and after conventional cancer therapy in head and neck SCC by database systematic search.

This is an attractive perspective because, in clinical practice, PD-L1 assessment directly affects the selection of cancer therapy.

We thank the Reviewer for His/Her interest in our work.

Point 1: However, it should be drastically updated for publication. 

This manuscript is partially difficult to read due to immaturity in English language expression and sentence structure. Specifically, the introduction is redundant, there is some overlap between the discussion and results parts, and there are errors in the refs (e.g., #8). 

Response 1: We thank the Reviewer for the suggestions. The manuscript has been edited by a native speaker. We checked the text throughout and rephrased parts of the Introduction to improve the English language and reduce amount of text and redundancies. We checked the references for mismatches, and reduced text redundancies between Results and Discussion.

Point 2: Fig 1 is very confusing to understand and needs a significant layout change. The order of authors' names is meaningless. It is preferable to list by treatment method or by clone. It would also be informative to specify the period between the start (or completion) of treatment and the next IHC.

Response 2: We thank the Reviewer for the suggestions. The Figure 1 is based on the official template of Preferred Reporting Items for Systematic Reviews and Meta-Analyses (PRISMA) statement which is recommended to be used. We worked on the Table to list the studies by IHC clone as suggested, and we added in the text the information on time between the start (or completion) of treatment and the next IHC, when it was available in the included studies.

Round 2

Reviewer 2 Report

Line 79; Is Ref 8 not the report of KEYNOTE-040 but CheckMate141?

The authors should clarify the meaning of ‘combined therapy’ in the entire manuscript. For example, in Table 2, Leduc et al. published a study of not combined chemo-radiotherapy but chemotherapy alone. If the author uses the term ‘combined’ as the ‘combined use of a different type of chemotherapy’, it should be clarified. And these 3 studies in Table 2 contain both TC and IC, so clarify them in Table 2.

Line 256; The study of So et al (ref 45) contains patients treated with chemotherapy (4/42), therefore, ‘the effect of radiotherapy alone’ seems not to be correct.

The discussion section is still redundant and confusing, although I can understand that it is difficult to conclude from published studies containing heterogeneity. Please consider revising the text to make it easier for the reader.

Author Response

Point 1: Line 79; Is Ref 8 not the report of KEYNOTE-040 but CheckMate141?

 Response 1: We thank the Reviewer for help us noticing. We corrected with the correct reference to KEYNOTE-040.

Point 2: The authors should clarify the meaning of ‘combined therapy’ in the entire manuscript. For example, in Table 2, Leduc et al. published a study of not combined chemo-radiotherapy but chemotherapy alone. If the author uses the term ‘combined’ as the ‘combined use of a different type of chemotherapy’, it should be clarified. And these 3 studies in Table 2 contain both TC and IC, so clarify them in Table 2.

Response 2: We thank the Reviewer for the comment. The three studies included in meta-analysis had all chemotherapy regimens comprising cisplatin and two of them also included radiotherapy (with Leduc et al. with a chemotherapy regimen only), and we intended the “combined therapy” as the use of both chemotherapy and radiotherapy. We checked the use of the terminology in the manuscript. Concerning TC and IC in these three studies, we reported correctly that Leduc et al. and Delafoy et al. scored both TC and IC, but Ock et al. scored only TC.

Point 3: Line 256; The study of So et al (ref 45) contains patients treated with chemotherapy (4/42), therefore, ‘the effect of radiotherapy alone’ seems not to be correct.

Response 3: We thank the Reviewer for help us noticing. We corrected the text accordingly, highlightig that in the paper of So et al. there were patients treated with chemotherapy but the results report only on the effect of radiotherapy.

Point 4: The discussion section is still redundant and confusing, although I can understand that it is difficult to conclude from published studies containing heterogeneity. Please consider revising the text to make it easier for the reader.

Response 4: We thank the Reviewer for the comment. We reduced parts of the discussion to remove redundancies and rephrased some parts to improve readalbility.